# Patterns of Diet, Physical Activity, Sitting and Sleep Are Associated with Socio-Demographic, Behavioural, and Health-Risk Indicators in Adults

**DOI:** 10.3390/ijerph16132375

**Published:** 2019-07-04

**Authors:** Stina Oftedal, Corneel Vandelanotte, Mitch J. Duncan

**Affiliations:** 1School of Medicine & Public Health; Faculty of Health and Medicine, The University of Newcastle, University Drive, Callaghan NSW 2308, Australia; 2Priority Research Centre for Physical Activity and Nutrition, The University of Newcastle, University Drive, Callaghan NSW 2308, Australia; 3Physical Activity Research Group, Appleton Institute, School of Health, Medical and Applied Sciences, Central Queensland University, Rockhampton Queensland 4702, Australia

**Keywords:** health behaviour, lifestyle, body mass index, mental health

## Abstract

Our understanding of how multiple health-behaviours co-occur is in its infancy. This study aimed to: (1) identify patterns of physical activity, diet, sitting, and sleep; and (2) examine the association between sociodemographic and health-risk indicators. Pooled data from annual cross-sectional telephone surveys of Australian adults (2015–2017, n = 3374, 51.4% women) were used. Participants self-reported physical activity, diet, sitting-time, sleep/rest insufficiency, sociodemographic characteristics, smoking, alcohol use, height and weight to calculate body mass index (BMI), and mental distress frequency. Latent class analysis identified health-behaviour classes. Latent class regression determined the associations between health-behaviour patterns, sociodemographic, and health-risk indicators. Three latent classes were identified. Relative to a ‘moderate lifestyle’ pattern (men: 43.2%, women: 38.1%), a ‘poor lifestyle’ pattern (men: 19.9%, women: 30.5%) was associated with increased odds of a younger age, smoking, BMI ≥ 30.0 kg/m2, frequent mental distress (men and women), non-partnered status (men only), a lower Socioeconomic Index for Areas centile, primary/secondary education only, and BMI = 25.0–29.9 kg/m^2^ (women only). An ‘active poor sleeper’ pattern (men: 37.0%, women: 31.4%) was associated with increased odds of a younger age (men and women), working and frequent mental distress (women only), relative to a ‘moderate lifestyle’ pattern. Better understanding of how health-behaviour patterns influence future health status is needed. Targeted interventions jointly addressing these behaviours are a public health priority.

## 1. Introduction

Sufficient physical activity, a high-quality diet, and an adequate amount of good quality sleep are independently associated with reduced all-cause mortality [1]. These behaviours co-occur, and the way an individual engages in one behaviour can influence engagement in other behaviours. For example, poor sleep is associated with poorer dietary quality [2]. Distinct patterns of individual health behaviours have been identified [3,4], such as the “weekend warrior” who performs all their exercise in 1–2 sessions a week [5]. However, the understanding of how multiple health behaviours, such as physical activity, sitting, diet, and sleep, co-occur within individuals is still in its infancy [6].

Engaging in a greater number of negative health behaviours has been associated with a greater mortality risk [1,7], while taking part in more positive health behaviours is associated with greater well-being [8]. An improved understanding of health-behaviour patterns can assist in better understanding how these behaviours in combination influence health and wellbeing. In terms of behaviour change, it is thought to be more efficacious to address multiple health behaviours simultaneously instead of through single-behaviour interventions [9]. However, in order to design relevant interventions that address multiple behaviours, a better understanding of patterns of physical activity, diet quality, sitting time, and sleep at the population level is necessary. Understanding how such patterns vary by socio-demographic, behavioural, and health-risk indicators, such as smoking, alcohol use, body mass index (BMI), and mental distress, is also of key importance, in order to identify priority groups for future interventions. 

Cassidy et al [10] demonstrated that relative to adults with a BMI of 18.5–29.9 kg/m^2^, adults with a BMI of 25.0–29.9 kg/m^2^ or ≥30 kg/m^2^ were more likely to report low levels of physical activity (≤967.5 Metabolic Equivalents of Tasks minutes/week), higher TV viewing (>3 hours/day), and either shorter (<7 hours/night) or longer (>8 hours/night) than recommended hours of sleep per night. Whilst useful, examining the likelihood of engaging in poor health behaviours does not provide information on the distinct patterns of these behaviours present, or how these patterns relate to weight status or other characteristics. One study, which explored how sleep and physical activity co-occur, found four distinct patterns [11]. The group whose behaviour was characterised by poor self-rated sleep quality and low physical activity (<150 min/week) had the highest prevalence of BMI > 30 kg/m^2^, poor/fair self-rated health, three or more chronic health conditions, and a higher level of psychological distress [11]. Studies that also include diet quality and sitting-time would add further information in terms of understanding health-risk [12]. For example, it has been shown that improvements in cardio-metabolic markers from addressing excessive sitting time are attenuated by partial sleep restriction [13]. 

The understanding of how activity, diet, and sleep behaviours co-occur is still evolving, and using data-driven approaches such as latent class analysis (LCA) to identify population sub-groups with similar behavioural patterns is a promising approach to population health [14]. By identifying behavioural patterns, it is possible to identify which key health-risk behaviours co-occur and examine descriptive correlates of the identified subgroups. This provides more comprehensive context to inform which behaviours to target and what measures may be efficacious when tailoring interventions to the groups of interest. 

Therefore, the aims of the current study were to: (1) examine patterns of physical activity, diet quality, sitting time, and sleep/rest insufficiency in adults; and (2) examine how these health-behaviour patterns are associated with socio-demographic, behavioural, and health-risk indicators.

## 2. Methods

The National Social Survey (NSS) is a cross-sectional survey conducted annually by the Population Research Laboratory at CQUniversity (CQU), using randomly selected mobile and landline numbers from an Australian database and random digit dialing of numbers from all Australian states and territories [15]. Eligible participants were aged 18 years and over, resided in Australia, gave verbal informed consent prior to the start of the interview, and provided complete data for all relevant variables (i.e., those with missing data were excluded). All data was collected over the phone. This study used data from the 2015, 2016, and 2017 NSSs. The CQUniversity Human Ethics Research Review Panel approved each NSS (H14/09-203). The authors had full access to all data.

### 2.1. Health Behaviour Measures 

The daily intake of fruit and vegetables and weekly frequency of fast food consumption was self-reported. Participants were asked to quantify their fruit and vegetable intake: “How many serves of vegetables do you eat on a usual day? One serve of vegetables is equivalent to half a cup of cooked vegetables or one cup of salad vegetables”; and ”How many serves of fruit do you eat on a usual day? One serve of fruit is equivalent to one medium piece or two small pieces of fruit”. Fruit intake was categorised as ‘≥2 serves/day’, ‘1–2 serves/day’, and ‘none’. Vegetable intake was categorised as ‘≥5 serves/day’, ‘1–4 serves/day’, or ‘none’. Having ≥2 serves of fruit and ≥5 serves of vegetables is seen as meeting Australian dietary guidelines [16]. For fast food, they were asked: “In the last week, how many times did you eat something from a fast-food restaurant like McDonald’s, Hungry Jacks, or KFC? This also includes other fast-food and takeaways, for example, fish and chips, Chinese food, and pizza”. Fast food intake was categorised as ‘never’, ‘once per week’, or ‘2–7 times per week’. 

Physical activity during the last seven days was assessed using the Active Australia Questionnaire [17]. The sum of time spent walking and doing moderate and vigorous physical activity (with vigorous activity weighted by two) gave total minutes of activity. The frequency of activity sessions performed for more than 10 min was reported. Participants were categorised as: ‘inactive (0 min/week and 0 sessions), ‘insufficiently active’ (1 to 149 min/week and less than five sessions), ‘sufficiently active’ (≥150 min/week and five sessions or more) [17].

Sitting time was assessed using two items from the International Physical Activity Questionnaire [18]. Total sitting time was calculated as the weighted sum of weekday and weekend sitting time and subsequently classified into three categories: sitting ‘<8 h per day’, ‘8–11 hours per day’, and ‘>11 hours per day’. This categorisation was chosen due to evidence showing that sitting for 8–11 h (Hazard Ratio (HR): 1.15, 95% CI: 1.06–1.25), and >11 h (HR: 1.40, 95% CI: 1.27–1.55) were associated with greater mortality risk compared to 4–8 h [19].

Participants were asked: “During the past 30 days, for about how many days have you felt you did not get enough rest or sleep?” Responses were categorised as none (0 days), infrequent (1–13 days), or frequent (14–30 days). This single item has demonstrated acceptable levels of test–retest reliability [20].

### 2.2. Socio-Demographic, Behavioural, and Health-Risk Indicators

Participants self-reported sex and age (years). The highest level of educational attainment was collapsed as ‘primary/secondary school’, ‘Technical and Further Education (TAFE)/technical college’, or ‘university’. Marital status was categorised as ‘single, widowed, separated, or divorced’ or ‘married or de-facto’. Work status was dichotomised as ‘in workforce’ or ‘not in workforce/retired’. Postcodes were used to assign each participant a Socio-economic Indexes for Areas (SEIFA) score using the Index of Relative Advantage and Disadvantage [21]. Participants reported if they lived in a ‘city’, ‘town’, or ‘rural’ geographical location. Smoking status was dichotomised as ‘current smoker’ or ‘non-smoker’. The Alcohol Use Disorders Identification Test—version C was used to measure alcohol consumption, and responses were dichotomised into ‘low-risk’ (0–2 standard drinks per drinking occasion) and ‘high-risk’ (>2 standard drinks per usual drinking occasion and more than 4 on any occasion) alcohol consumption, in line with Australian guidelines [22]. Participants’ self-reported height and weight were used to calculate body mass index (BMI), which was subsequently categorised as <18.5 kg/m^2^, 18.5–24.9 kg/m^2^, 25.0–29.9 kg/m^2^, or ≥30 kg/m^2^. The frequency of mental distress was assessed using a single item from the Behavioural Risk Factor Surveillance System ‘Healthy Days Module’: “Now, thinking about your mental health, which includes stress, depression, and problems with emotions, for how many days during the last 30 days was your mental health not good?” [23].

### 2.3. Statistical Analysis

Latent class analysis (LCA) was used to identify health behaviour classes. A series of LCA, specifying two to six classes, were examined to explore the underlying structure of the six behavioural indicators, in order to identify the fewest number of classes that best represented combinations of the behaviours. One hundred iterations of each model (i.e., from two to six classes) were performed using randomly generated seed values and were compared using G^2^ criterion values. The Bayesian and Akaike Information Criterions (BIC and AIC) were generated for each LCA, where a lower BIC and AIC suggest better goodness of fit. These data were used in combination with the interpretability of the solution to select the appropriate number of classes and maximise model fit. Separate models were examined for men and women by estimating and comparing the G^2^ criterion of models, constrained and unconstrained by sex, using the selected three-class solution. The G^2^ criterion values significantly differed (*p* < 0.05), confirming that estimating models separately for men and women significantly improved model fit. The relationships between class membership, socio-demographic, behavioural, and health-risk covariates were assessed using a latent class regression approach [24]. The continuous SEIFA centile and frequency of mental distress were transformed to z-scores prior to inclusion in analysis so that a one-unit change in the covariate was equivalent to a change of one standard deviation in the covariate (SEIFA-centile: 1 SD = 2.8 deciles, frequent mental distress: 1 SD = 7.1 days). Age was continuous (years) and all other covariates were categorical, as described in the methods. Due to the small sample of participants with BMI < 18.5kg/m^2^ (n = 67, 2%), they were grouped with the 18.5–24.9kg/m^2^ category as the reference group in the regression. Using an inclusive method, including all covariates of interest, each participant was assigned to a class via the maximum-probability approach to describe the sociodemographic, behavioural, and health-risk indicators of the latent classes [25]. Because the classification of individuals to latent classes is a form of imputation, each individual has a probability of membership in each latent class [25]. This is known as posterior probabilities, and the maximum-probability approach assigns a participant to the class for which they have the highest probability of membership [25]. The PROC LCA command procedure in SAS version 9.4 (Release 9.4. SAS Institute Inc., Cary, NC, USA) was used to estimate model parameters. Stata (v12.0, StataCorp, College Station, TX, USA) was used for data management and ANOVA analysis to describe the study population.

## 3. Results

The response rates for the 2015, 2016, and 2017 NSSs were 33%, 26%, and 24%, respectively, similar to other telephone surveys [26], which provided 3800 responses. The final study sample with complete data (n = 3374, 88.8%) consisted of 1640 men (48.6%) and 1734 women (51.4%), with a mean (SD) age of 53.1 (17.8) years, 66.7% were partnered and 46.1% had a university education (Table 1).

A three-class solution provided the best fit and most interpretable solution. Item-response probabilities for each class were significantly different for men and women, but the underlying patterns were similar, and the same names were used to characterise the classes using the most differentiating behaviours found in the response probability data (Table 2) for each class. Figure 1 shows the probability for recommended fruit and vegetable intake, no fast food, recommended activity level, ≤8 h/day of sitting time, and no days of insufficient sleep or rest for each behaviour class for men and women. All classes had a high probability of sitting for less than eight hours a day but varied on other behaviours: The ‘moderate lifestyle’ class (43.2% of men and 38.1% of women) had the highest probability of eating ≥2 serves of fruit a day, not eating fast food, and reporting zero days of insufficient sleep or rest per month. They had the highest probability of consuming ≥5 serves of vegetables per day and the second highest probability of meeting physical activity recommendations;The ‘active poor sleepers’ class (37.0% of men and 31.4% of women) had the highest probability of meeting physical activity recommendations. Their fruit and vegetable intake was comparable to that of the ‘moderate lifestyle class’ but a third of men (33%) and a fifth of women (18%) ate fast food 2–7 times per week. They had the highest probability of 1–14 days of insufficient sleep or rest in the last month;The ‘poor lifestyle’ class (19.9% of men and 30.5% of women) had the highest probability of having no serves of fruit or vegetables per day and reporting 14–30 days of insufficient sleep or rest in the last month. Fast-food consumption frequency was similar to that of the ‘active poor sleepers’. This class was most likely to be insufficiently active or inactive.

Using latent class regression (Table 3), relative to the ‘moderate lifestyle’ class, men and women in the ‘active poor sleepers’ class were found to have lower odds of being older. Women in the ‘active poor sleeper’ class also had lower odds of being retired or not in the workforce and higher odds of reporting more frequent mental distress (Table 3). Both men and women in the ‘poor lifestyle’ class had lower odds of being older, higher odds of being a smoker, and report more frequent mental distress. Men in the ‘poor lifestyle’ class had higher odds of not having a partner and a BMI ≥ 30 kg/m^2^. Women in the ‘poor lifestyle’ class had higher odds of having a primary or secondary education only, belonging to a lower SEIFA centile, and having a BMI between 25–29.9 kg/m^2^ or ≥30 kg/m^2^ (Table 3). Alcohol use was not different between classes, nor was geographical location. 

Each class is characterised in Table 4. Of note is the stepped increase in the frequency of mental distress between the classes. The ‘moderate lifestyle’ class reports, on average, less than one day of mental distress in the last 30 days (men: 0.6 ± 2.3 days; women: 0.5 ± 1.6 days), the ‘active poor sleeper class’ reporting approximately three days (men: 2.7 ± 5.6 days; women: 3.3 ± 5.9 days), and the ‘poor lifestyle’ class reporting nine to ten days (men: 10.5 ± 12.0 days; women: 8.8 ± 11.1 days). Of those reporting ≥15 days of mental distress in the last 30 days, three quarters (men: 72.9%, women: 79.8%) were in the poor lifestyle class (Table 4). 

## 4. Discussion

This study identified three population clusters, which differed in prevalence and probability of engaging in physical activity, sitting, diet, and sleep for men and women, but fit under the same broad class descriptions of ‘moderate lifestyle’, ‘active poor sleepers’, and ‘poor lifestyle’. The behaviour patterns of men and women within the classes were comparable, with the greatest differences observed for the dietary measures. This is similar to other Australian surveys, where women were more likely than men to consume fruits and vegetables in line with national recommendations [27]. The ‘poor lifestyle’ behaviour class, which described 19.9% of men and 30.5% of women, was characterised by a pattern of physical inactivity, poor diet, and insufficient sleep or rest that increased chronic disease risk. They also had sociodemographic and health-risk factor correlates that differed to the other classes identified in the current study. There was no significant difference in geographical location (city, town, or rural) between the classes, and while 48.8% of the study sample reported high-risk alcohol consumption, this health-risk behaviour did not significantly differ between classes.

Those in the ‘poor lifestyle’ class were more likely to be younger, current smokers, have a higher weight status, and report more frequent mental distress, and men were less likely to have a partner. Women were also more likely to have primary or secondary school as their highest completed education and more likely to live in an area ascribed a lower SEIFA decile, which is consistent with studies demonstrating that ‘riskier’ health-behaviour classes are more likely to have a lower socioeconomic profile [28]. Improvements in health behaviours have happened at a faster pace in the least disadvantaged subpopulations in Australia compared to the most disadvantaged populations, creating a widening gap over time [29]. Given the potential for synergistic relationships between activity, diet, and sleep, improving multiple lifestyle behaviours concurrently may have the potential to provide greater health benefits relative to improvements in a single behaviour. However, producing interventions that effectively improve health behaviours in lower socioeconomic groups requires careful consideration of social inequalities, and acknowledgement of the fact that participation in positive health behaviours is not an entirely free choice [30].

Men and women in the ‘poor lifestyle’ class reported experiencing mental distress more frequently in the last month. This is in agreement with other studies, which found that frequent mental distress is associated with a higher prevalence of poor diet and inactivity [31]. However, women in the ‘active poor sleepers’ class also reported more frequent mental distress, despite having higher levels of physical activity than the ‘moderate lifestyle’ class. The significant link between poor sleep and mental distress might explain this association [32]. Other studies exploring patterns of health behaviours that include sleep and mental distress have reported similar findings [6]. Health behaviours appear to exert synergistic effects on health [1] and insufficient sleep or rest may negate some of the positive effects of health-promoting behaviours, such as physical activity and diet. Furthermore, subjective poor sleep quality and higher emotional stress have been found to be significant predictors of incident BMI ≥ 30 kg/m^2^ at follow-up (mean time: 7.5 years), with an additive effect between the two [33]. While the mechanism behind this finding is still unknown, this could partly explain the increased prevalence of higher BMI in the ‘poor lifestyle’ class. This class has the highest probability of insufficient sleep or rest for ≥14 days in the last 30 days (Table 2), and the highest odds of frequent mental distress in the last 30 days (Table 3). 

Targeting individuals with a higher weight status for health behaviour change is frequently encouraged, using dietary restriction, improved diet quality, and increased physical activity as a means of attempting weight-loss [34]. However, significant improvements in health outcomes are available via adopting health behaviours independent of weight status or weight change [35,36,37]. Moreover, it is important to note that only 12.8% (n = 93) of men and 25.9% of women (n = 127) with a BMI of 25.0–29.9 and 36.9% of men (n = 131) and 55.9% of women (n = 237) with a BMI ≥ 30.0 were characterised by a ‘poor lifestyle’ pattern (Table 4). It is therefore important to view weight status in combination with health behaviours to identify individuals with a higher risk of poor health outcomes, rather than weight status alone. 

Inadequate sleep is a significant public health issue in Australia, and while partly related to clinical sleep disorders and other health complaints, a significant portion is due to work or lifestyle-related sleep restriction [38]. National surveys have found that 33–45% of Australians complain of insufficient sleep on a daily or multiple days per week basis, and this includes the approximate 11.3% of Australians who have insomnia [38]. In the current overall study population, 40.9% and 25.5% reported insufficient sleep or rest 1–13 and 14–30 days per week, respectively, leaving only one-third (33.6%) who reported zero days of insufficient sleep or rest in the last 30 days (Table 1). Insufficient sleep was also prevalent in two behaviour classes with otherwise different behaviour profiles (‘active poor sleepers’ and ‘poor lifestyle’). Despite this, sleep health has not received the same attention as a preventative health behaviour as physical activity and diet quality. The costs from the loss of well-being associated with poor sleep amounts to an estimated US$27.33 billion [38]. Research suggests that even for those who report insomnia—defined as sleep initiation or maintenance problems plus daytime consequences (sleepiness, fatigue or exhaustion, irritability or moodiness) at least three times a week, despite adequate opportunity and circumstances for sleep—only a small proportion (5–13%) seek help [39]. Barriers include trivialisation of the problem, believing it will resolve spontaneously, lack of awareness of available treatments, and the perception that treatment is ineffective or unattractive [39]. Traditional treatments for sleep disorders are also very costly, as they are primarily delivered face-to-face, and alternative strategies are needed [40]. Public education on the importance of sleep health, increased awareness of how sleep health interrelates with other health behaviours, and easily accessible, wide-reaching interventions integrating multiple behaviours are required. 

A significant strength of the study was the focus on multiple behaviours that influence health and wellbeing in an interrelated manner. The inclusion of mental distress was also a particular strength due to its strong association with sleep and other health-risk behaviours [32,33]. Including mobile phone numbers increased the representativeness of the sample [41]. The findings must, however, be seen in light of the study’s limitations, which include its cross-sectional nature, and as such no causation or directionality can be determined from the data. Another limitation was the use of self-report for all measures, which may have contributed to participants being misclassified due to their over or under reporting their behaviours. Furthermore, the questions used related to a relatively short period, limiting our ability to ascertain long-term habits. 

## 5. Conclusions

This study identified three distinct patterns of physical activity, diet quality, sitting, and sleep that were differentially associated with sociodemographic characteristics, smoking, weight status, and mental distress, but not alcohol use. Poor sleep was prevalent in two out of three behaviour patterns and appeared to be associated with increased mental distress. While the health benefits of sufficient physical activity, a high-quality diet, and an adequate amount of good quality sleep are well researched, a better understanding of how these behaviours interact to influence health and how to improve these behaviours is a public health priority.

## Figures and Tables

**Figure 1 ijerph-16-02375-f001:**
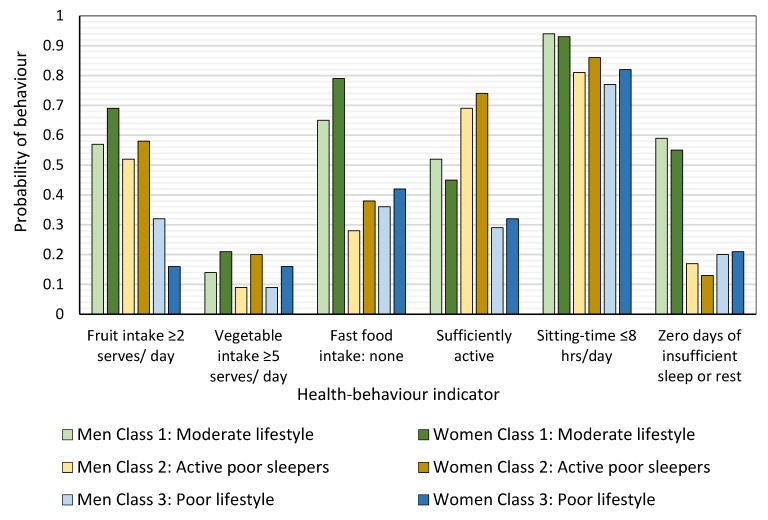
Item-response probability plot for men and women by health behavior class. The probability of each health-behaviour in each of the three behaviour classes ‘moderate lifestyle’, ‘active poor sleeper’, and ‘poor lifestyle’ (alternating bars for men and women).

**Table 1 ijerph-16-02375-t001:** Study population characteristics (n = 3374).

		Male (n = 1640)	Female (n = 1734)	Total (n = 3374)
		**Mean (SD)**		
Age	Years	52.9 (18.2)	53.2 (17.4)	53.1 (17.8)
SEIFA decile	Out of 10	6.2 (2.8)	6.1 (2.8)	6.1 (2.8)
Mental distress	Days per last 30 days	3.1 (7.2)	3.8 (7.7)	3.5 (7.5)
		**Count (%)**		
Marital status	Partnered	1129 (69.2)	1114 (64.4)	2243 (66.7)
	Single, widowed, divorced	503 (30.8)	615 (35.6)	1118 (33.3)
Education	University	732 (44.6)	823 (47.5)	1555 (46.1)
	TAFE or trade college	387 (23.6)	353 (20.4)	740 (21.9)
	Primary/Secondary	521 (31.8)	558 (32.2)	1079 (32.0)
Work status	Currently working	1028 (62.7)	954 (55.0)	1982 (58.7)
	Not in workforce/ retired	612 (37.3)	780 (45.0)	1392 (41.3)
Geographical location	City	878 (53.5)	875 (50.5)	1753 (52.0)
	Town	349 (21.3)	403 (23.2)	752 (22.3)
	Rural	413 (25.2)	456 (26.3)	869 (25.8)
Smoking	Current smoker	238 (14.5)	212 (12.2)	450 (13.3)
	Non-smoker	1402 (85.5)	1522 (87.8)	2924 (86.7)
Alcohol	High risk drinking	849 (51.8)	799 (46.1)	1648 (48.8)
	Low risk drinking	791 (48.2)	935 (53.9)	1726 (51.2)
Body mass index ^a^	<18.5 kg/m^2^	17 (1.0)	50 (2.9)	67 (2.0)
	18.5–24.9 kg/m^2^	541 (33.0)	769 (44.4)	1310 (38.8)
	25.0–29.9 kg/m^2^	727 (44.3)	491 (28.3)	1218 (36.1)
	≥30 kg/m^2^	355 (21.7)	424 (24.5)	779 (23.1)
Fruit intake	None	229 (14.0)	196 (11.3)	425 (12.6)
	1 serve/ day	585 (35.7)	549 (31.7)	1134 (33.6)
	≥2 serves/ day	825 (50.3)	989 (57.0)	1814 (53.8)
Vegetable intake	None	47 (2.9)	27 (1.6)	74 (2.2)
	1–4 serves/ day	1411 (86.1)	1370 (79.1)	2781 (82.5)
	≥5 serves/ day	181 (11.0)	336 (19.4)	517 (15.3)
Fast food frequency	2–7 times/week	354 (21.6)	205 (11.8)	559 (16.6)
	1 times/week	541 (33.0)	572 (33.0)	1113 (33.0)
	Never	744 (45.4)	956 (55.2)	1700 (50.4)
Physical activity level ^b^	Inactive	191 (11.7)	218 (12.6)	409 (12.2)
	Insufficiently active	559 (34.3)	656 (38.0)	1215 (36.2)
	Sufficiently active	881 (54.0)	853 (49.4)	1734 (51.6)
Sitting-time	>11 hrs/day	70 (4.3)	61 (3.5)	131 (3.9)
	8–11 hrs/day	166 (10.1)	157 (9.1)	323 (9.6)
	≤8 hrs/day	1404 (85.6)	1516 (87.4)	2920 (86.5)
Insufficient sleep in last 30 days	14–30 days	358 (21.9)	499 (28.8)	857 (25.5)
	1–13 days	697 (42.6)	681 (39.3)	1378 (40.9)
	None/ zero	581 (35.5)	551 (31.8)	1132 (33.6)

SEIFA: Socio-Economic Indexes for Areas; ^a^ BMI < 18.5 kg/m^2^ and 18.5–24.9 kg/m^2^ grouped in regression analysis; ^b^ ‘inactive’ = 0 min/week and 0 sessions; ‘insufficiently active’ = 1 to 149 min/week and less than five sessions; ‘sufficiently active’ ≥150 min/week and five sessions or more.

**Table 2 ijerph-16-02375-t002:** Latent class analysis by sex, item-response probabilities for health behaviour indicators (n = 3374).

		Men (n = 1640)			Women (n = 1734)		
		Moderate Lifestyle	Active Poor Sleepers	Poor Lifestyle	Moderate Lifestyle	Active Poor Sleepers	Poor Lifestyle
Latent class membership (%)		43.2	37.0	19.9	38.1	31.4	30.5
Fruit intake	None	0.07	0.09	0.40	0.06	0.05	0.23
	1 serve/ day	0.36	0.39	0.28	0.25	0.36	0.36
	≥2 serves/ day	0.57	0.52	0.32	0.69	0.58	0.16
Vegetable intake	None	0.01	0.01	0.12	0.00	0.00	0.05
	1–4 serves/ day	0.86	0.90	0.79	0.79	0.80	0.79
	≥5 serves/ day	0.14	0.09	0.09	0.21	0.20	0.16
Fast food frequency	2–7 times/week	0.06	0.33	0.32	0.01	0.18	0.19
	1 time/week	0.28	0.39	0.32	0.20	0.45	0.39
	Never	0.65	0.28	0.36	0.79	0.38	0.42
Physical activity level	Inactive	0.13	0.03	0.28	0.13	0.01	0.24
	Insufficiently active	0.36	0.28	0.43	0.43	0.25	0.44
	Sufficiently active	0.52	0.69	0.29	0.45	0.74	0.32
Sitting-time	>11 hrs/day	0.02	0.04	0.10	0.02	0.01	0.08
	8–11 hrs/day	0.04	0.15	0.14	0.13	0.13	0.10
	≤8 hrs/day	0.94	0.81	0.77	0.93	0.86	0.82
Insufficient sleep or rest in last 30 days	14–30 days	0.11	0.19	0.54	0.11	0.27	0.53
	1–13 days	0.30	0.64	0.26	0.24	0.60	0.25
	None/zero	0.59	0.17	0.20	0.55	0.13	0.21

**Table 3 ijerph-16-02375-t003:** Odds of class membership by sociodemographic and health-risk characteristics (n = 3374).

	Men (n = 1644)			Women (n = 1737)		
	Moderate Lifestyle OR (95%CI)	Active Poor Sleepers OR (95%CI)	Poor Lifestyle OR (95%CI)	Moderate LifestyleOR (95%CI)	Active Poor SleepersOR (95%CI)	Poor Lifestyle OR (95%CI)
Age ^a^	Ref	0.88 (0.85–0.91)	0.92 (0.89–0.96)	Ref	0.88 (0.84–0.91)	0.88 (0.85–0.92)
Single, widowed or divorced ^b^	Ref	1.31 (0.55–3.13)	2.13 (1.06–4.29)	Ref	1.84 (0.81–4.20)	1.67 (0.78–3.61)
Primary or secondary school education only ^c^	Ref	0.57 (0.25–1.35)	1.88 (0.84–4.20)	Ref	0.84 (0.33–2.12)	2.98 (1.29–6.88)
TAFE or trade college education ^c^	Ref	0.80 (0.32–1.99)	1.95 (0.87–4.37)	Ref	0.95 (0.38–2.39)	2.01 (0.84–4.81)
Retired or not in workforce ^d^	Ref	0.37 (0.13–1.05)	1.00 (0.48–2.08)	Ref	0.37 (0.16–0.91)	0.76 (0.36–1.59)
SEIFA score (z-score) ^e^	Ref	1.23 (0.83–1.84)	0.91 (0.64–1.30)	Ref	1.14 (0.77–1.70)	0.70 (0.49–0.99)
Geographic location: town ^f^	Ref	0.57 (0.22–1.42)	0.57 (0.26–1.26)	Ref	1.13 (0.48–2.64)	0.84 (0.37–1.89)
Geographic location: rural ^f^	Ref	0.81 (0.33–1.97)	0.65 (0.29–1.47)	Ref	0.67 (0.27–1.69)	0.76 (0.33–1.77)
Current smoker ^g^	Ref	0.83 (0.29–2.38)	5.12 (2.15–12.19)	Ref	0.90 (0.29–2.73)	3.46 (1.22–9.78)
High-risk alcohol consumption ^h^	Ref	1.45 (0.67–3.17)	0.87 (0.45–1.67)	Ref	1.42 (0.70–2.88)	0.90 (0.46–1.75)
BMI 25–29.9 kg/m^2 i^	Ref	1.19 (0.52–2.66)	1.41 (0.63–3.14)	Ref	1.54 (0.68–3.48)	2.61 (1.11–6.16)
BMI ≥ 30 kg/m^2 i^	Ref	0.66 (0.21–2.01)	4.81 (1.91–12.12)	Ref	1.21 (0.43–3.41)	8.16 (3.38–19.69)
Frequency of mental health distress (z-score) ^j^	Ref	2.43 (0.76–7.72)	4.73 (1.49–15.05)	Ref	8.29 (1.86–36.84)	11.88 (2.69–46.69)

^a^ Mean ± SD age: 53.3 ± 18.1 years; ^b^ compared to partnered; ^c^ compared to university; ^d^ compared to working; ^e^ mean ± SD Socioeconomic Indexes For Areas score: 6.2 ± 2.8; ^f^ compared to geographic location ‘city’; ^g^ compared to ‘non-smoker’; ^h^ compared to low-risk alcohol consumption; ^i^ compared to BMI < 25.0 kg/m^2^; ^j^ mean ± SD days of mental distress: 2.4 ± 7.1 days.

**Table 4 ijerph-16-02375-t004:** Distribution of sociodemographic, behavioural, and health-risk indicators by latent class (n = 3374).

	Men			Women		
	Moderate Lifestyle (n = 726)	Active Poor Sleepers (n = 614)	Poor Lifestyle (n = 292)	Moderate Lifestyle (n = 687)	Active Poor Sleepers (n = 540)	Poor Lifestyle (n = 502)
Age, mean (SD)	67.1 (10.2)	37.0 (12.8)	51.2 (14.2)	67.2 (10.4)	39.3 (12.9)	49.2 (14.8)
Mental distress, mean (SD)	0.6 (2.3)	2.7 (5.6)	10.5 (12.0)	0.5 (1.6)	3.3 (5.9)	8.8 (11.1)
SEIFA decile, mean (SD)	5.8 (2.7)	6.9 (2.7)	5.5 (2.7)	6.0 (2.8)	7.1 (2.6)	5.1 (2.7)
	*n (%)*	*n (%)*	*n (%)*	*n (%)*	*n (%)*	*n (%)*
Marital status						
Married or de-factor	580 (51.4)	390 (34.5)	159 (14.1)	476 (42.7)	330 (29.6)	308 (27.7)
Single, divorced, widowed	146 (28.0)	224 (44.5)	133 (26.4)	210 (34.2)	210 (34.3)	194 (31.5)
Education						
Primary or secondary school	296 (56.8)	99 (19.0)	126 (24.2)	277 (49.6)	57 (10.2)	224 (40.1)
TAFE or trade college	168 (43.4)	118 (30.5)	101 (26.1)	147 (41.6)	75 (21.3)	131 (37.1)
University degree	266 (36.4)	400 (54.9)	66 (9.0)	265 (32.2)	409 (49.7)	149 (18.1)
Geographical location						
City	333 (37.9)	395 (45.0)	150 (17.1)	327 (37.4)	330 (37.7)	218 (24.9)
Town	168 (48.1)	114 (32.7)	67 (19.2)	162 (40.2)	121 (30.0)	120 (29.8)
Rural	229 (55.9)	108 (26.2)	76 (18.4)	200 (43.9)	90 (19.7)	166 (36.4)
Smoking status						
Non-smoker	679 (48.4)	543 (38.7)	180 (12.8)	651 (42.8)	510 (33.5)	361 (23.7)
Current smoker	51 (21.4)	74 (31.1)	113 (47.5)	38 (17.9)	31 (14.6)	143 (67.5)
Alcohol use						
Low risk alcohol use	358 (45.3)	278 (35.2)	155 (19.6)	392 (41.9)	237 (25.4)	306 (32.7)
High risk alcohol use	372 (43.8)	339 (39.9)	138 (16.3)	297 (37.2)	304 (38.1)	198 (24.8)
BMI category						
<18.5 kg/m^2^	6 (35.3)	8 (47.1)	3 (17.7)	20 (40.0)	23 (46.0)	7 (14.0)
18.5–25 kg/m^2^	206 (38.0)	269 (49.7)	66 (12.2)	301 (39.1)	335 (43.6)	133 (17.3)
25.0–29.9 kg/m^2^	368 (50.6)	266 (36.6)	93 (12.8)	229 (46.6)	135 (27.5)	127 (25.9)
≥30 kg/m^2^	150 (42.3)	74 (20.9)	131 (36.9)	139 (32.8)	48 (11.3)	237 (55.9)
Mental distress (in last 30 days)						
None/zero	638 (57.1)	361 (32.3)	119 (10.6)	573 (55.3)	263 (25.4)	201 (19.4)
1–13 days	87 (23.5)	220 (59.3)	64 (17.3)	115 (22.0)	244 (46.6)	165 (31.5)
14–30 days	5 (3.3)	36 (23.8)	110 (72.9)	1 (0.6)	34 (19.7)	138 (79.8)

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
