# Peer review of "Patterns of Diet, Physical Activity, Sitting and Sleep Are Associated with Socio-Demographic, Behavioural, and Health-Risk Indicators in Adults"

_ijerph, 2019, doi:10.3390/ijerph16132375_

Round 1

Reviewer 1 Report

Thank you for the opportunity to review manuscript ijerph-512582 titled "Patterns of Diet, Physical Activity, Sitting and Sleep Are Associated with Socio-Demographic, Behavioural and Health-Risk Indicators in Adults". This study identifies 3 distinct patterns of lifestyle and how they relate to sociodemographic characteristics and health risk-indicators. The manuscript is well written but some minor comments need to be addressed:

- line 27 "BMI 30.0" please add "≥" and the units "kg/m2"

- line 111 "Postcodes were 110 used to assign each participant a Socio-economic Indexes for Areas (SEIFA) score." Please clarify which index was used? SEIFA has 4 indexes:

        - the Index of Relative Socio-economic Disadvantage (IRSD)

        - the Index of Relative Socio-economic Advantage and Disadvantage (IRSAD)

        - the Index of Economic Resources (IER)

        - the Index of Education and Occupation (IEO)

Also, please clarify where the data was found (add a reference). I am assuming it was from the ABS for SEIFA 2016. 

- do the authors know whether the data was collected from people residing in large cities only or included people in small towns and rural areas? or in different states? If so, these could be sociodemographic variables worth including in the model as well.

- Table 1, the first variable "Sex" should be "n" or "number of participants"

- It is interesting to note that none of the participants were underweight (BMI<18.5 kg/m2)

- Figures 1 and 2, I feel the line plots are not easily read and suggest changing the plots to bar graphs. Also please include a y axis with tick marks to allow extraction of data if needed for meta analyses.

- Table 3, please clarify that numbers between the brackets are 95%CI

- Reference 12 should be updated to: Vincent, G.E.; Jay, S.M.; Sargent, C.; Kovac, K.; Vandelanotte, C.; Ridgers, N.D.; Ferguson, S.A. The impact of breaking up prolonged sitting on glucose metabolism and cognitive function when sleep is restricted. Neurobiol Sleep Circadian Rhythms 2018, 4, 17-23

- Reference 25 should be updated to: Ding, D.; Do, A.; Schmidt, H.-M.; Bauman, A.E. A Widening Gap? Changes in Multiple Lifestyle Risk Behaviours by Socioeconomic Status in New South Wales, Australia, 2002–2012. PLoS ONE 2015, 10(8): e0135338

Author Response

1.       Line 27 "BMI 30.0" please add "≥" and the units "kg/m2"

-          Author response: This has been added

2.       Line 111 "Postcodes were used to assign each participant a Socio-economic Indexes for Areas (SEIFA) score." Please clarify which index was used? SEIFA has 4 indexes

-          Author response: The Index of Relative Socio-economic Advantage and Disadvantage (IRSAD) was used and this has been added to the text

3.       Also, please clarify where the data was found (add a reference). I am assuming it was from the ABS for SEIFA 2016. 

-          Author response: A reference has been added (ABS for SEIFA 2011)

4.       Do the authors know whether the data was collected from people residing in large cities only or included people in small towns and rural areas? or in different states? If so, these could be sociodemographic variables worth including in the model as well.

-          Author response: The sampling methodology recruited participants from all states and territories as noted in the methods, and this included both metropolitan, regional and remote areas. Data on the participants self-reported geographic location was collected as living in a city, town or rural location. While we agree that this variable may be of interest, as we did not consider it a priori in the analysis we are somewhat reluctant to include it post hoc. We have noted in the limitations that geographic location was not examined. Another limitation was the use of self-report and the questions used related to a relatively short period, limiting our ability to ascertain long-term habits. The study also did not examine how patterns varied by geographic location or urbanisation.However, if the editor considers it essential we are open to re-analysing the data and revising the manuscript.

-          The distribution was as follows:

o   City: n=1753, 51.8%

o   Town: n=752, 22.2%

o   Rural: n=869, 25.7%

o   Missing: n=7, 0.2%

5.       Table 1, the first variable "Sex" should be "n" or "number of participants"

-          Author response: This has been corrected

6.       It is interesting to note that none of the participants were underweight (BMI<18.5 kg/m2)

-          Author response: There were indeed participants with a BMI of <18kg/m2, though not many (n=67 [17 men, 50 women], 2% of total sample). Because of this they were not included separately in the multivariate analysis but rather grouped with the 18.5-24.9 group. This omission from the methods has been addressed as below and also in the table.  Table 1 and 4 now include the BMI<18.5 kg/m2, and the reference group for Table 3 has been corrected to <25.0 kg/m2.

-          Page 4, line 155-56 now states: “Due to a small sample of participants with BMI <18.5kg/m2 (n=67, 2%), they were grouped with the 18.5-24.9kg/m2 as the reference group in the regression.”

7.       Figures 1 and 2, I feel the line plots are not easily read and suggest changing the plots to bar graphs.

-          Author response: This has been changed.

8.       Also please include a y axis with tick marks to allow extraction of data if needed for meta-analyses.

-          Author response: This has been included.

9.       Table 3, please clarify that numbers between the brackets are 95%CI

-          Author response: This has been added

10.   Reference 12 should be updated to: Vincent, G.E.; Jay, S.M.; Sargent, C.; Kovac, K.; Vandelanotte, C.; Ridgers, N.D.; Ferguson, S.A. The impact of breaking up prolonged sitting on glucose metabolism and cognitive function when sleep is restricted. Neurobiol Sleep Circadian Rhythms 2018, 4, 17-23

-          Author response: This has been updated

11.   Reference 25 should be updated to: Ding, D.; Do, A.; Schmidt, H.-M.; Bauman, A.E. A Widening Gap? Changes in Multiple Lifestyle Risk Behaviours by Socioeconomic Status in New South Wales, Australia, 2002–2012. PLoS ONE 2015, 10(8): e0135338

-          Author response: This has been updated.

Reviewer 2 Report

This paper is novel in assessing the co-existence of multiple health behaviors including sleep, diet and physical activity on sociodemographic and health risk indicators. The paper may be improved by addressing the following:

Is there an individual measure of income available?

The authors should provide more details on the analysis-specifically a brief description of the maximum-probability approach to assign participants to a class

The authors should test for trend across the categories of mental distress for class groups

The authors should discuss the figures in the text, and the minimum provide a detailed footnote explaining the figure. 

The authors discuss insomnia in the Discussion, but the authors included a measure of “..did not get enough rest or sleep”, this may be a proxy for sleepiness, or short sleep duration or insomnia. The authors should introduce in insomnia and explain/discuss why they believe the insufficient sleep measured used in the study is likely a proxy for insomnia. 

The authors did not adjust for stress or anxiety which could confound the results, were these variables available? 

The authors should proofread the paper 

Author Response

1.       Is there an individual measure of income available?

-          Yes, this was collected, but as the majority (67%) of the sample was partnered we used household income as this is the most common economic indicator of disposable income and standard of living.

2.       The authors should provide more details on the analysis-specifically a brief description of the maximum-probability approach to assign participants to a class.

-          Author response: This was added: Because the classification of individuals to latent classes is a form of imputation, each individual has a probability of membership in each latent class  [24]. This is known as posterior probabilities, and the maximum-probability approach assigns a participant to the class for which they have the highest probability of membership [24].”

3.       The authors should test for trend across the categories of mental distress for class groups

-          Author response: We are unsure of which results/ table the reviewer is referring to. If it is table 3, the mental distress outcome is continuous. In table 4 we have used both continuous and categories, but as the behaviour classes are not ordinal, we are unsure if testing for a trend is suitable.

4.       The authors should discuss the figures in the text, and the minimum provide a detailed footnote explaining the figure. 

-          Author response: This has been included.

5.       The authors discuss insomnia in the Discussion, but the authors included a measure of “..did not get enough rest or sleep”, this may be a proxy for sleepiness, or short sleep duration or insomnia. The authors should introduce in insomnia and explain/discuss why they believe the insufficient sleep measured used in the study is likely a proxy for insomnia. 

-          Author response: This section of the discussion has been modified to introduce insomnia. We did not necessarily mean that it is a proxy measure, but rather wanted to highlight that even those with insufficient sleep at the level of meeting the criteria for insomnia do not tend to seek help. The section now reads: “Inadequate sleep is a significant public health issue in Australia, and while partly related to clinical sleep disorders and other health complaints, a significant portion is due to work or lifestyle-related sleep restriction [34]. National surveys have found 33-45% of Australians complain of insufficient sleep on a daily or multiple days per week basis, and this includes the approximate 11.3% of Australians who have insomnia [34]. In the current study population, 41.0% and 25.5% reported insufficient sleep or rest 1-13 and 15-30 days per week respectively, leaving only one-third (33.6%) who reported no insufficient sleep or rest in the last 30 days (Table 1). Despite this, sleep health has not received the same attention as a preventative health behaviour as physical activity and diet quality. The costs from loss of well-being associated with poor sleep amounts to an estimated US$27.33 billion [34]. Research suggests that even for those who report insomnia, defined as sleep initiation or maintenance problems plus daytime consequences (sleepiness, fatigue or exhaustion, irritability or moodiness) at least three times a week despite adequate opportunity and circumstances for sleep only a small proportion (5-13%) seek help [34, 35]. Barriers include trivialisation of the problem, believing it will resolve spontaneously, lack of awareness of available treatments and the perception that treatment is ineffective or unattractive [38]. Traditional treatments for sleep disorders are also very costly as they are primarily delivered face-to-face, and alternative strategies are needed [39].

6.       The authors did not adjust for stress or anxiety which could confound the results, were these variables available? 

-          Author response: We did not adjust for stress or anxiety as this is captured within the ‘mental distress’ measure as a ‘catch-all’ for poor mental health (wording of question as per page 3, line 119-121): ‘now thinking about your mental health, which includes stress, depression and problems with emotions, for how many days during the last 30 days was your mental health not good?’).  Ref: David G. Moriarty, Matthew M. Zack, James B. Holt, Daniel P. Chapman, Marc A. Safran, Geographic Patterns of Frequent Mental Distress: U.S. Adults, 1993–2001 and 2003–2006, Am J Prev Med, 2009: 36 (6): 497-505.

7.       The authors should proofread the paper.

-          Author response:  This has been done.

Reviewer 3 Report

This is an interesting paper illustrating a LCA approach to 'profile' health behaviours in an Australian population sample.  More details are needed in the methods to describe how variables were measured.

Introduction:

 Line 58: it would be good to include duration values here.

 Line 61: reference needed

Methods:

2.1

·     Needs to be clear that this survey was verbally conducted over the phone.

·     The actual questions asked to determine fruit and vegetable intake should be stated.  

·     What guidelines were given as to a ‘serving’?  Were fruit/vegetable juices included?

·     Rationale for ‘sitting hours’ cut off to be provided (tertiles ?)

·     Sleep question – please state the actual question asked to participants 

2.2 

AUDIT-C / BRESS – are these acronyms? If so please write in full.

2.3 

Were there any missing data points? If so how were they dealt with?

Results:

Table 1 ‘work hours’ would read better as ‘work status’

Figure 1 y-axis legend I think should be ‘probability of behaviour’

Discussion:

 Be clear that the ‘sleep’ variable also included ‘rest’ therefore hard to fully interpret.

Higher BMI and frequent mental distress are potentially linked (lines 31) - need to expand this point.

Limitations - please expand re lack of validation of questions against objective measures (if they have been validated against object measures then please state).

Author Response

1.       Author response: Line 58: it would be good to include duration values here.

-          Duration has been added: “Cassidy et al [10] demonstrated that relative to adults with a BMI of 18.5-29.9 kg/m2, adults with a BMI of 25.0-29.9kg/m2 or ≥30kg/m2 have been found to be more likely to report low levels of physical activity (≤967.5 MET minutes/week), higher TV-viewing (>3 hours/day), and either shorter (<7 hours/night) or longer (>8 hours/night) than recommended hours of sleep per night. “

2.       Line 61: reference needed

-          Author response: This has been added.

3.       Needs to be clear that this survey was verbally conducted over the phone.

-          Author response: This has been included in methods.

4.       The actual questions asked to determine fruit and vegetable intake should be stated.  

-          Author response: This has been included in methods: “Participants were asked to quantify their fruit and vegetable intake: “How many serves of vegetables do you eat on a usual day? One serve of vegetables is equivalent to half a cup of cooked vegetables or one cup of salad vegetables” and” How many serves of fruit do you eat on a usual day? One serve of fruit is equivalent to one medium piece or two small pieces of fruit.

5.       What guidelines were given as to a ‘serving’?  Were fruit/vegetable juices included?

-          Author response: This has been included in methods (as above excerpt).

6.       Rationale for ‘sitting hours’ cut off to be provided (tertiles ?)

-          Author response: The following has been added:” This categorisation was chosen due to evidence showing that sitting for 8-11 hours (HR: 1.15, 95%CI: 1.06-1.25) and >11 hours (HR: 1.40,95%CI: 1.27-1.55) were associated with greater mortality risk compared to 4-8 hours [18].”

7.       Sleep question – please state the actual question asked to participants 

-          Author response: This has been added: “Participants were asked: “During the past 30 days, for about how many days have you felt you did not get enough rest or sleep?””

8.       AUDIT-C / BRESS – are these acronyms? If so please write in full.

-          Author response: This has been added (Alcohol Use Disorders Identification Tool version C, Behavioual Risk Factor Surveillance System)

9.       Were there any missing data points? If so how were they dealt with?

-          Author response: Those with missing data were excluded. We mentioned in the results that 3800 surveys were collected where 3381 had complete data for our variables, however we have added to the methods section that those with missing data were excluded to make this clearer.

10.   Table 1 ‘work hours’ would read better as ‘work status’

-          Author response: Thank you, his has been changed

11.   Figure 1 y-axis legend I think should be ‘probability of behaviour’

-          Author response: Yes, this has been corrected.

12.   Be clear that the ‘sleep’ variable also included ‘rest’ therefore hard to fully interpret.

-          Author response: We have included ‘rest’ when discussing this item throughout.

13.   Higher BMI and frequent mental distress are potentially linked (lines 31) - need to expand this point.

-          Author response: The paper referred to found emotional distress predicted incidence BMI>+30kg/m2 at follow-up, and this was expanded on so it is clearer: “A longitudinal study of short sleep duration and incidence BMI≥30kg/m2 in individuals with a BMI <30kg/m2  at baseline, found subjective poor sleep quality and higher emotional stress were significant predictors of incident BMI≥30kg/m2 at follow up, with an additive effect between the two [31]. While the mechanism behind this finding is still unknown, this could partly explain the increased prevalence of higher BMI in the ‘poor lifestyle’ class…”

14.   Limitations - please expand re lack of validation of questions against objective measures (if they have been validated against object measures then please state).

-          Author response: This was added: Another limitation was the use of self-report for all measures, which may have contributed to participants being misclassified due to their over or under reporting their behaviours“

Round 2

Reviewer 2 Report

The authors have greatly improved their manuscript by addressing the comments. 

Author Response

Thank you very much for reviewing our paper.

Kind regards,

Stina Oftedal